# Religious Attendance in a Secular Country Protects Adolescents from Health-Risk Behavior Only in Combination with Participation in Church Activities

**DOI:** 10.3390/ijerph17249372

**Published:** 2020-12-15

**Authors:** Marie Buchtova, Klara Malinakova, Alice Kosarkova, Vit Husek, Jitse P. van Dijk, Peter Tavel

**Affiliations:** 1Olomouc University Social Health Institute, Palacký University Olomouc, 771 11 Olomouc, Czech Republic; klara.malinakova@oushi.upol.cz (K.M.); alice.kosarkova@oushi.upol.cz (A.K.); vit.husek@upol.cz (V.H.); j.p.van.dijk@umcg.nl (J.P.v.D.); peter.tavel@oushi.upol.cz (P.T.); 2Department of Community and Occupational Medicine, University Medical Center Groningen, University of Groningen, 9713 AV Groningen, The Netherlands; 3Graduate School Kosice Institute for Society and Health, P.J. Safarik University in Kosice, 040 11 Kosice, Slovakia

**Keywords:** adolescents, religiosity, spirituality, health-risk behavior, HBSC study

## Abstract

Religiosity and spirituality have been considered to be protective factors of adolescent health-risk behavior (HRB). The aim of this study was to assess the relationship between adolescents’ HRB and their religiosity, taking into account their parents’ faith and their own participation in church activities. A nationally representative sample (*n* = 13377, 13.5 ± 1.7 years, 49.1% boys) of Czech adolescents participated in the 2018 Health Behavior in School-aged Children cross-sectional study. We measured religious attendance (RA), faith importance (FI) (both of respondents and their parents), participation in church activities and adolescent HRB (tobacco, alcohol, and cannabis use and early sexual intercourse). We found that neither RA nor FI of participants or their parents had a significant effect on adolescents’ HRB. Compared to attending respondents who participate in church activities (AP), non-attending respondents who participate in church activities were more likely to report smoking and early sexual intercourse, with odds ratios (ORs) ranging from 3.14 (1.54–6.39) to 3.82 (1.99–7.35). Compared to AP, non-attending respondents who did not participate in church activities were more likely to report early sexual intercourse, with OR = 1.90 (1.14–3.17). Thus, our findings show that RA does not protect adolescents from HRB; they suggest that RA protects adolescents from HRB only in combination with participation in church activities.

## 1. Introduction

Adolescent health-risk behavior (HRB) is the subject of many research studies worldwide. Research shows that it may be a predictor of adult risk behavior [1,2,3,4,5]. Although the numbers in the Czech Republic are decreasing, according to data from the Czech 2018 Health Behavior in School-aged Children (HBSC) study, adolescents still have a high prevalence of tobacco and marijuana use: 41% of fifteen-years old girls and 37% of fifteen-years old boys have smoked tobacco at least once in their lives, and 21% of girls and 16% of boys in this age group have smoked during the last 30 days. Furthermore, 20% of boys and 17% of girls have already tried cannabis. In regular marijuana consumption, Czech adolescents hold 17th place out of 42 surveyed countries with 8% of boys and 7% of girls who have regular contact with marijuana. Moreover, 76% of girls and about the same number of boys have drunk alcohol [6]. Because of these high numbers based on the data from 45 countries that participated in the 2017/2018 HBSC survey, it is necessary to search for ways to protect young people from these risks.

Religiosity and spirituality (R/S) are often considered as protective factors of risk behavior [7]. They may help to postpone the onset of smoking or reduce its occurrence among adolescents [8,9]. Moreover, R/S can also act as a protective factor of sexual behavior [10,11], and religious attendance (RA) is associated with a lower alcohol and substance use [12,13,14,15]. However, some research confirmed gender differences in the relationship between R/S and HRB [9,15], and some studies [16] have not found a link between spirituality and substance use among adolescents. An explanation might be that adolescent risk behavior is influenced by a lot of factors. In terms of adolescents’ smoking, the influence of their parents and peers is significant, and adolescents tend to start smoking when their parents and peers smoke [17,18]. An adolescents’ decision to start smoking is also related to their personal faith practice and belonging to a religious community [8]. Furthermore, Kim-Spoon et al. found an inverse correlation between adolescent substance use and their parents’ religiosity [19]. It is possible that religiosity is associated with a greater parental control [20] and higher self-control in adolescents, thus reducing the tendency to substance abuse [19]. In the same way, the influence of a religious peer group could be a possible protective factor in adolescent risk behavior. Some sources, e.g., Bartkowski et al. [21], also take into account the influence of a faith community on human development, suggesting that positive human development may be more strongly influenced by a religious community than by family.

According to earlier studies, the Czech Republic was considered one of the most secular or even atheistic countries in the world [22,23], with a high percentage of religiously unaffiliated people [24,25]. The church attendance rate is quite low in the Czech republic, only 8% citizens go to church regularly at least once a month [26]. Compared to other European nations, the position of religion in the Czech Republic has significantly weakened [22] and Czechs are characterized by the weakest relationship to the Christian church as an institution. However, Christianity itself is not perceived so negatively [23], and so contemporary Czech society can be described as post-Christian rather than secular or atheistic. It appears that family education is still a key factor in the continuity of religious life in secular Czech society [27]. Research has thus far not focused much on the transmission of faith and the influence of a religious community on adolescents’ behavior. Therefore, the aim of this article is to analyze the relationship of health-risk behavior and adolescents’ religious attendance and faith in a secular environment, taking the faith of their parents and a background among believing/religious peers into account.

## 2. Materials and Methods

### 2.1. Participants and Procedure

A nationally representative sample of Czech boys and girls was obtained through the 2018 Health Behavior in School-aged Children (HBSC) study. This cross-sectional WHO collaborative study focused on health and health-related behavior and their socioeconomic determinants in 11-, 13-, and 15-year-old children. The HBSC study has been conducted at 4-year intervals since 1983/84 and now includes 48 countries across Europe and North America. According to the HBSC study protocol, schools were selected randomly after stratification by region, school size, and type of school (primary schools vs. secondary schools). Out of 227 contacted schools, 7 refused to participate and were substituted by schools of similar size in their close neighborhood; thus, the school response rate was 97%. Then, classes from the 5th, 7th, and 9th grades, in general corresponding to age the categories of 11-, 13-, and 15-year-olds, were selected at random, one from each grade per school. Data from 13,885 pupils were obtained (response rate 86.4%). The majority of non-response was due to illness (8.87%) or other reasons for absence (4.18%); however, 84 children refused to participate in the survey (0.5%). Of the sample, 238 questionnaires were excluded due to the problems found during the check of internal consistency and due to missing responses to key questions, and another 270 respondents were excluded because their age lay outside the age-range of the specific class. Thus, 13,377 questionnaires were considered valid.

Data was collected between May and June 2018 and was gathered through online surveys. Instructions were given by trained administrators with no teachers present in the classroom in order to reduce response bias. Each participant received a unique application code from the administrator to access to the questionnaire. Respondents had one school lesson (45 min) to complete the questionnaire. Participation in the survey was anonymous and voluntary. All survey procedures for each data collection cycle have been saved and can be downloaded [28]. The Czech HBSC study was conducted under the auspices of the Ministry of Education, Youth and Sports of the Czech Republic and the World Health Organization Country Office in the Czech Republic. The study design was approved by the Ethics Committee of the Faculty of Physical Culture, Palacký University Olomouc (No. 9/2016).

### 2.2. Measures

“Religious attendance” was measured by the question: “How often do you go to church or to religious sessions?” with possible answers: several times a week/once a week/once a month/several times a year/rarely/not at all. Weekly attendance is compulsory for most Czech religions [29], so participants who reported at least one weekly religious meeting were dichotomized as attending, as used in previous studies based on HBSC data [15]. Parental religious attendance was measured in the same way by asking questions similar to those asked of adolescents: “How often does your father/mother go to church or to religious sessions?” and was dichotomized in the same way as adolescents’ religious attendance.

“Importance of faith” was measured by the question: “How important is faith in God for you?” Possible answers were on a scale from 1 (not at all) through 4 (neither important nor unimportant) to 7 (absolutely). According to the previously used dichotomization [15], participants responding from 5 to 7 on the scale were dichotomized as those for whom faith is important. The importance of faith for parents was examined in the same way by the question: “In your view, how important is faith in God for your father/mother?” and was dichotomized in the same way.

“Participation in church activities” was assessed by the question: “In your free time, do you take part in any religious activities (e.g., church meetings, singing in church choirs)?” The possible answers were “Yes” or “No”.

“Smoking” was assessed by the question: “How often do you currently smoke tobacco?” with possible answers: every day/at least once a week but not daily/less often than once a week/I don’t smoke. According to the HBSC dichotomization [30,31], respondents who smoked at least once a week were classified as smokers, the rest as non-smokers. Moreover, this dichotomization has been used for Czech adolescents before [32].

“Drinking” was measured by the question: “How many days (if ever) have you drunk alcohol in the last 30 days?” with possible answers: never/1–2 days/3–5 days/6–9 days/10–19 days/20–29 days/30 days or more. Respondents were classified as alcohol consumers if they reported drinking at least three days in the last month, as used in previous studies based on HBSC data [32,33,34].

“Recent cannabis use” was assessed by the question: “Have you used cannabis (weed, ganja) in the last 30 days?” with the answers: never/1–2 days/3–5 days/6–9 days/10–19 days/20–29 days/30 days or more. As in some previous studies [35,36], those who answered “never” or “1–2 days” were classified as cannabis non-users, the rest of the respondents as users.

“Early sexual intercourse” was measured by the question: “Have you ever had sexual intercourse (sometimes this is called “making love”, “having sex”, etc.)?” Possible answers were “Yes” or “No”.

“Age, gender, socioeconomic status and social media” use were considered potentially confounding variables and were obtained by the questionnaire. The socioeconomic status of the respondents’ families was assessed using the Family Affluence Scale (FAS). This scale examines the number of cars in the family, having one’s own bedroom, the number of computers (including laptops and tablets) in the household, the number of bathrooms, dishwasher ownership, and the number of family holidays abroad last year. The summary score ranges from 10 to 13 and, according to HBSC recommendations, we transformed ordinal data into an interval scale with a normalized range (from 0 to 1, with a higher score indicating higher socioeconomic position) and distribution.

Social media use of the respondents was assessed using the 9-item Social Media Disorder Scale [37] measuring social media addiction degree. The wording of the questions was: “We are interested in your experiences with social media. The term social media refers to social network sites (e.g., Facebook, Instagram, Twitter) and instant messengers (e.g., WhatsApp, Snapchat, Facebook messenger). During the past year, have you … “followed by nine disorder characteristics (e.g., regularly found that you can’t think of anything else but the moment that you will be able to use social media again?). Possible answers to each item were “Yes” or “No”. We performed a summary score of all “Yes” answers.

### 2.3. Statistical Analyses

As a first step, we described the background characteristics of the sample. The normality of the data was verified using the Shapiro–Wilk test. Since the data was not normally distributed, non-parametric methods were used for the statistical analyses. Then, we assessed the associations of various health-risk behaviors with religious attendance (Model 1) and the importance of faith (Model 2) for both the respondents themselves and their parents using binary logistic regression models. Each model was first tested as a crude one and was consequently adjusted for gender, age, socioeconomic status, and social media use. In the same way we used a binary logistic model to assess the associations between health-risk behaviors and different combinations of religious attendance and church activities. All analyses were performed using the statistical software package IBM SPSS version 25 (New York, NY, USA).

## 3. Results

### 3.1. Description of the Population

The background characteristics of the sample are presented in Table 1, which also describes the prevalence of the four types of health-risk behavior for the whole sample as well as for the sample divided according to RA and according to importance of faith. Of the whole sample, 5.9% respondents attended church services at least weekly and 17.8% reported that faith is personally important for them. Regarding RA and importance of faith in parents, 6% of respondents reported that their mothers attended church at least once a week (5.2% for fathers), and 20% (respectively 17.2%) of adolescents reported faith is important for their mothers (respectively fathers). The adolescents’ own RA was highly correlated with their mothers’ RA (Spearman’s *r* = 0.84 (*p* < 0.01)) and their fathers’ RA (*r* = 0.79, *p* < 0.01). Importance of faith for the respondents was also highly correlated with their parents’ importance of faith (with *r* = 0.85 for mothers and *r* = 0.82 for fathers, respectively).

### 3.2. Health-Risk Behavior

Table 2 shows the associations of religious attendance and importance of faith with various health-risk behaviors, adjusted for age, gender, socioeconomic status, and social media use. We found almost no significant association of health-risk behavior and church attendance (Model 1) or the importance of faith (Model 2), either in terms of respondents or of their parents. However, we found a significant association between early sexual intercourse and respondents’ fathers church attendance. Specifically, respondents whose fathers did not attend church were 1.6 times more likely to report early sexual intercourse.

### 3.3. Church Activities

Table 3 presents the associations of identical health-risk behaviors with different combinations of religious attendance and church activities, adjusted for age, gender, socioeconomic status, and social media use. Respondents who reported religious attendance but did not participate in any church activity were more likely to use alcohol, although the result was significant only for the crude model (a 72% increase in the odds) and not for the adjusted one. However, for a combination of religious non-attendance with participation in church activities, the odds ratios were significantly increased for smoking, drinking, and early sexual intercourse, although the results for cannabis use were significant for the crude model only. Respondents who neither attended religious meetings nor participated in church activities were 1.90 times more likely to report early sexual intercourse. See Figure 1 for graphical representation.

## 4. Discussion

The aim of this study was to assess the relationship between adolescent health-risk behavior and their religiosity, also taking into account their parents’ faith and participation in church activities. The results show that neither church attendance nor the importance of faith in adolescents’ personal life had a significant effect on their health-risk behavior. However, a combination of religious attendance and participation in church activities was associated with generally lower adolescent health-risk behavior. Adolescents who did not attend church but participated in church activities at the same time turned out to be the most vulnerable group in terms of risk behavior. This group was found to have a significantly higher chance for all measured types of health-risk behavior. Respondents who did not attend church or religious communities regularly had a higher chance of early sexual intercourse.

We found that religious attendance or the importance of faith in the personal life of adolescents alone have almost no significant impact on their health-risk behavior. Moreover, the only significant relationship emerging from our results does not relate directly to the religiosity of the participants, but to the religiosity of their fathers. These findings differ from the results of some previous studies, which reported an inverse relationship between adolescent R/S and substance use [12,13,14,38] or early sexual intercourse [11,39]. Many recent studies have shown that R/S reduces alcohol consumption [40,41,42,43,44] or delays the onset of alcohol abuse in adolescents [45]. However, some studies have come to different conclusions. Pokhrel et al. [16] reported no significant associations between spirituality and substance use. Moreover, according to Hannauer et al. [46], the importance of religion significantly increases young people’s perceptions of the risk of alcohol abuse and cannabis use, but has no effect on the perceived risk of smoking. When we focus on the European environment, the results of recent similar research suggest that the relationship between religiosity and risky behavior in adolescents also differs by gender [9,15] and regions [15,16,47,48]. In contrast to the results of [47] or [14], our study did not confirm the inverse relationship between the religiosity and alcohol consumption and cannabis use in adolescents, respectively.

The different conclusions may be caused by different methodological approach. As multinomial constructs, both religiosity and spirituality are difficult to define, and they often overlap in the scientific literature [38]. Furthermore, a different methodological approach can lead to a different definition of variables and measurement. To compare, Good and Willoughby [12] discuss institutional (e.g., church attendance and presence at religious gatherings) and personal (e.g., personal experience with the sacred, and the frequency of personal prayer) R/S.

Furthermore, we found that participation in regular church worship is perceived as an external manifestation of R/S and the subjective importance of faith in personal life as personal R/S. Moreover, it should also be noted that a secular environment can play a role, as the Czech Republic is considered one of the most secular countries in the world [25]. In contrast, most studies on adolescent R/S and its impact on health-risk behavior have been conducted in predominantly religious countries [12,19,49]. The results of most research, therefore, agree with the well-established view of the inverse relationship between spirituality and risky behavior, which, however, does not correspond to our findings.

We found that religious attendance was a protective factor for health-risk behavior when combined with participation in church activities. Compared to these respondents, the adolescents who only participated in church activities without religious attendance were more likely to report higher health-risk behaviors. Thus, our results are in accordance with the findings of Malinakova et al. [29], who reported that mere church attendance has only a limited impact on adolescent health-risk behavior. Their study also showed the importance of internalization of religious values in order to protect adolescents from risk behavior. In our case, non-attending respondents who participate in church activities can represent adolescents with internal spiritual needs who do not consider regular church attendance as a form of fulfilment. Thus, adolescents’ church non-attendance can be seen as a form of rejection of a formal and somehow compulsory duty. Such refusal of standards may be further manifested by increased tendency for risk behavior. In addition, our results, highlighting the importance of internalizing religious values are consistent with some recent findings [50,51] that discrepancies in the religious values of parents and adolescents lead to higher levels of HRB.

In our study, religious attendance was protective for adolescent health-risk behavior only when combined with participation in other church activities. Thus, our results confirm the conclusions of many studies [52,53,54] that the influence of peer groups is crucial in terms of risky behavior of adolescents. They extend the findings of other studies reporting a protective role of R/S [7,10,12] by also stressing the importance of sharing religious values with peers, especially in a secular environment. However, it is obvious that religious activities alone, without regular church attendance, are not sufficient for protecting adolescents from health-risk behavior.

### 4.1. Strengths and Limitations

The main strength of our study is its large and representative sample size, together with a high response rate. Another strength is the use of the established HBSC methodology. However, the relatively small number of respondents who attend church regularly and consider religious faith important can be seen as a limitation, which may have decreased the power of the study, especially when combined with participation in religious activities. A further limitation can be informational bias, as our data are self-referenced and this can be influenced by social desirability. The last limitation is the cross-sectional study design, which does not allow us to draw conclusions on causalities.

### 4.2. Implications

Our study highlights the importance of the influence of peers and informal groups on the health-risk behavior of adolescents. In addition to regular church attendance and the importance of faith for personal life, it is the peer groups that can encourage non-risky behavior. Our findings may be important for people working with children and young people in the field of education, social, or pastoral care. Particularly in pastoral work, various church communities may play an effective role in the prevention of adolescent health-risk. Workers in this area should also be informed that non-attending respondents who participate in church activities might represent a more vulnerable group.

Our results also show that due to the ambiguity of the definitions of religiosity and spirituality there is no universal question to measure them. Especially in a secular country, the influence of the environment needs to be taken into account. Further research is needed for a deeper understanding of the function of peer groups within church communities and of their impact on risk behavior, especially their behavior as a subculture.

## 5. Conclusions

We found that mere religious attendance or the perceived importance of faith have a negligible impact on adolescent health-risk behavior. Religious attendance was found to be protective, but only in combination with participation in church activities. In contrast, the most vulnerable group of respondents were those who did not attend church regularly but participated in church-organized activities. The supportive role of church communities and religious peer groups in terms of prevention of health-risk behavior can be a subject of further research.

## Figures and Tables

**Figure 1 ijerph-17-09372-f001:**
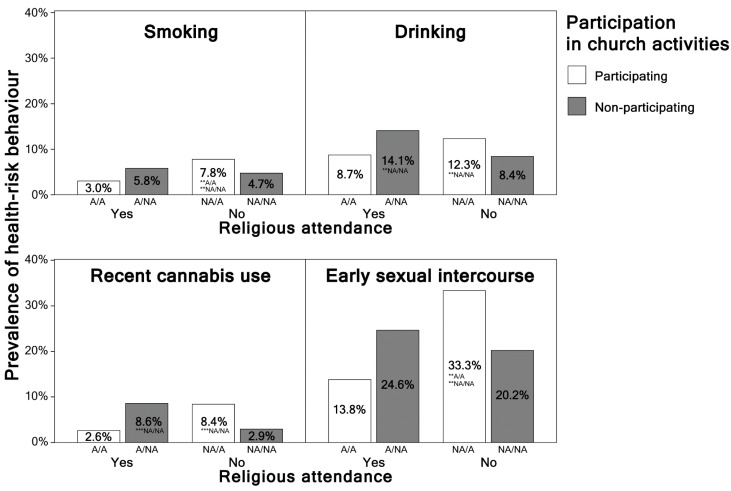
Prevalence of adolescent smoking, drinking, recent cannabis use, and early sexual intercourse in groups with different combinations of religious attendance and participation in church activities (Czech Republic, 2018). (Notes: ** *p* < 0.01, *** *p* < 0.001; A/A = attending/church activities; A/NA = attending/no church activities; NA/A = non-attending/church activities; NA/NA = non-attending/no church activities).

**Table 1 ijerph-17-09372-t001:** Sample characteristics by religious attendance and importance of faith.

	Number	%	Religious Attendance				*p*-Value	Importance of Faith				*p*-Value
			Attending		Non-Attending			Important		Non-Important		
			Number	%	Number	%		Number	%	Number	%	
Gender							*n.s.*					*n.s.*
Boys	6808	50.9	371	51.1	5817	50.7		1079	49.7	5100	50.9	
Girls	6569	49.1	355	48.9	5661	49.3		1091	50.3	4921	49.1	
Age							*p* < 0.05					*p* < 0.001
11 years old (5th grade)	4380	32.7	254	35.0	3491	30.4		851	39.2	2895	28.9	
13 years old (7th grade)	4654	34.8	245	33.7	4068	35.4		715	32.9	3593	35.9	
15 years old (9th grade)	4343	32.5	227	31.3	3919	34.1		604	27.8	3533	35.3	
Health-risk behavior ^a^												
Smoking	637	4.8	28	3.9	553	4.9	*n.s.*	100	4.6	484	4.9	*n.s.*
Drinking	1116	8.5	73	10.2	969	8.6	*n.s.*	172	8.0	873	8.9	*n.s.*
Recent cannabis use (9th grade only)	136	3.2	10	4.4	121	3.1	*n.s.*	14	2.3	115	3.3	*n.s.*
Early sexual intercourse (9th grade only)	855	20.3	40	17.9	794	20.6	*n.s.*	119	20.1	711	20.4	*n.s.*
Parental religiosity												
Mother’s church attendance	722	6.0	581	82.1	131	1.2	*p* < 0.001	498	23.5	211	2.1	*p* < 0.001
Father’s church attendance	626	5.2	488	69.7	127	1.1	*p* < 0.001	438	20.8	177	1.8	*p* < 0.001
Mother’s importance of faith	2412	20.0	576	80.8	1816	16.1	*p* < 0.001	1782	83.5	628	6.3	*p* < 0.001
Father’s importance of faith	2075	17.2	493	69.7	1563	13.8	*p* < 0.001	500	5.0	1572	73.9	*p* < 0.001
Church activities	1060	8.1	498	70.2	455	4.1	*p* < 0.001	585	27.7	366	3.7	*p* < 0.001
Total	13,377	100	726	5.9	11478	94.1		2170	17.8	10,021	82.2	

Notes: Number of missing cases per variable: Smoking—104; Drinking—4943; Recent cannabis use—9089; Early sexual intercourse—9167; Religious attendance—1173; Importance of faith—1186; Mother’s church attendance—1257; Father’s church attendance—1306; Mother’s importance of faith—1299; Father’s importance of faith—1316; Church activities—301. ^a^ Only numbers regarding the respondents with the occurrence of health-risk behavior are presented.

**Table 2 ijerph-17-09372-t002:** Associations of adolescent smoking, drinking, recent cannabis use, and early sexual intercourse with religious attendance, importance of faith, crude and, adjusted for age, gender, socioeconomic status (FAS), and social media use (odds ratios and 95% confidence intervals).

		Smoking		Drinking		Recent Cannabis Use (15 Years Old)		Early Sexual Intercourse (15 Years Old)	
Model 1: Religious Attendance		Crude	Adjusted	Crude	Adjusted	Crude	Adjusted	Crude	Adjusted
Respondent	Non-attending vs. attending	1.26 (0.85–1.85)	1.25 (0.82–1.90)	0.83 (0.64–1.06)	0.78 (0.59–1.03)	0.69 (0.36–1.34)	0.65 (0.33–1.27)	1.18 (0.83–1.68)	1.46 (1.00–2.16)
Mother	Non-attending vs. attending	1.23 (0.84–1.81)	1.17 (0.77–1.78)	0.83 (0.64–1.06)	0.81 (0.61–1.07)	0.76 (0.38–1.53)	0.72 (0.36–1.45)	0.95 (0.68–1.32)	1.14 (0.79–1.65)
Father	Non–attending vs. attending	1.06 (0.72–1.56)	1.08 (0.71–1.65)	0.98 (0.74–1.31)	1.02 (0.74–1.39)	0.58 (0.30–1.12)	0.55 (0.28–1.07)	1.24 (0.85–1.80)	1.61 (1.05–2.46) *
Model 2: Importance of faith									
Respondent	Unimportant vs. important	1.05 (0.84–1.31)	0.90 (0.71–1.15)	1.11 (0.94–1.32)	0.96 (0.80–1.16)	1.42 (0.81–2.49)	1.87 (0.97–3.62)	1.02 (0.82–1.27)	1.07 (0.85–1.35)
Mother	Unimportant vs. important	1.22 (0.98–1.53)	1.07 (0.84–1.36)	1.04 (0.89–1.22)	0.90 (0.76–1.07)	1.21 (0.74–1.98)	1.22 (0.72–2.05)	0.96 (0.79–1.18)	0.96 (0.78–1.19)
Father	Unimportant vs. important	1.00 (0.80–1.26)	0.84 (0.66–1.07)	1.19 (0.99–1.42)	1.02 (0.84–1.24)	1.13 (0.66–1.92)	1.26 (0.70–2.28)	0.94 (0.75–1.16)	0.95 (0.75–1.20)

Notes: * *p* < 0.05.

**Table 3 ijerph-17-09372-t003:** Associations of adolescent smoking, drinking, recent cannabis use, and early sexual intercourse with different combinations of religious attendance with church activities, crude and, adjusted for age, gender, socioeconomic status (FAS), and social media use (odds ratios and 95% confidence intervals).

	Smoking		Drinking		Recent Cannabis Use (15 Years Old)		Early Sexual Intercourse (15 Years Old)	
	Crude	Adjusted	Crude	Adjusted	Crude	Adjusted	Crude	Adjusted
Attendance + Church activities	1 **	1 **	1 **	1 **	1 **	1	1 **	1 **
Attendance + No church activities	1.97 (0.91–4.28)	2.26 (0.97–5.26)	1.72 (1.04–2.83) *	1.32 (0.76–2.29)	3.47 (0.95–12.71)	3.65 (0.98–13.60)	2.04 (1.00–4.17)	2.05 (0.93–4.52)
Non-attendance + Church activities	2.71 (1.46–5.02) **	3.14 (1.54–6.39) **	1.47 (0.97–2.24)	1.38 (0.87–2.21)	3.39 (1.04–11.11) *	1.44 (0.35–5.95)	3.12 (1.72–5.67) ***	3.82 (1.99–7.35) ***
Non-attendance + No church activities	1.59 (0.94-2.68)	1.74 (0.96–3.14)	0.96 (0.70–1.33)	0.84 (0.60–1.19)	1.12 (0.41–3.08)	1.13 (0.41–3.13)	1.58 (1.00–2.52)	1.90 (1.14–3.17) *

Notes: * *p* < 0.05, ** *p* < 0.01, *** *p* < 0.001.

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
