# Peer review of "Religious Attendance in a Secular Country Protects Adolescents from Health-Risk Behavior Only in Combination with Participation in Church Activities"

_ijerph, 2020, doi:10.3390/ijerph17249372_

Round 1

Reviewer 1 Report

I enjoyed this article and praise the care and detail provided along with challenging results that compel attention.

ln 44 'high' compared to what? Without comparison this comment is baseless and unhelpful

ln 70-2 excellent statement of research objective

ln 74-136 sample well described. Except greater clarity on administration of on-line questionnaire in classes would be helpful. Measures are identified and described clearly. I dislike arbitrarily dichotomised variables as they obscure diversity and nuance. They are bare minimum, but adequate to a rough estimation of the relationships under investigation. Some idea of known parameters would be helpful - e.g. national rate of church attendance, - but again not needed to estimate the relationships in question.

ln 137 ff  solid statistical analysis

Clear tabular presentation of findings and well written summaries of key findings

 Findings are distinctive and highly reliable. They are well discussed and compared with other studies with useful discussion about potential sources of difference, without defensiveness.

Reviewer 2 Report

This was a sound article carried out according to well-established procedures. Its major points of interest were in the analytical discussion pertaining to possible contextual causation for the results, since they were somewhat contradictory to expectation in relation to other studies on similar topics.

The only thing that caused a pause in my reading was the use of the word 'worst' in line 65, which seemed unnecessarily judgmental. Probably 'weakest' relationship would be more accurate?

Reviewer 3 Report

The article provides interesting data but it lacks consistency in terms of relating the study case to previous research. It is a good example of a particular case but without connections to other similar studies carried out in other places, which makes it out of the academia track.

The authors are not claryfing which church do they intend to refer. Catholic Church? Other Christian Churches? Those assumptions are not academic enough.

The paper is a well-informed article on the behaviour of youth but it does not refer to youth studies. It also lacks social media factor: do these young people do what they do also because of social media pressure?

There is a missed connection to european surveys on youth and health.

The article in this stage can not be published but with major revisions it could benefit from scientific soundness.

Some references that could help: 

    • Alonso, Andoni, and Pedro Oiarzabal, eds. 2010. Diasporas in the New Media Age: Identity, Politics, and Community, 151168RenoUniversity of Nevada Press. [Google Scholar]
    • Appadurai, Arjun. 1996. Modernity at Large: Cultural Dimensions of Globalization. MinneapolisUniversity of Minnesota Press. [Google Scholar

  • Benson, Peter L.Michael J. Donahue, and Joseph A. Erickson1989. “Adolescence and Religion: A Review of the Literature from 1970 to 1986.” In Research in the Social Scientific Study of Religion: A Research Annual, edited by M. L. Lynn and D. O. Moberg, Vol. 1, 153181GreenwichJAI Press. [Google Scholar]
  • Benson, Peter L., and Michael J. Donahue1995“Religion and the Well-Being of Adolescents.” Journal of Social Issues 51: 145160. DOI: 10.1111/j.1540-4560.1995.tb01328.x. [Crossref], [Web of Science ®], [Google Scholar]
  • Benson, Peter L.Kevin S. Masters, and David B. Larson1997“Religious Influences on Child and Adolescent Development.” In Handbook of Child and Adolescent Psychiatry, Vol. 4, Varieties of Development, edited by N. E. Alessi206219New YorkJohn Wiley and Sons. [Google Scholar]
  • Campbell, Heidi A. 2013. Digital Religion, Understanding Religious Practice in New Media Worlds. LondonRoutledge. [Google Scholar]
  • Deaux, Kay. 1993“Reconstructing Social Identity.” Personality and Social Psychology Bulletin 19: 412. DOI: 10.1177/0146167293191001. [Crossref], [Web of Science ®], [Google Scholar]
  • Elkind, David. 1964“Age Changes in the Meaning of Religious Identity.” Review of Religious Research 6: 3640. DOI: 10.2307/3510881. [Crossref], [Web of Science ®], [Google Scholar]
  • Elkind, David. 1970. Children and Adolescents: Interpretative Essays on Jean Piaget. New YorkOxford University Press. [Google Scholar]
  • Elzo, Javier. 2005. Los jóvenes y la felicidad [Youth and Happiness]MadridEdiciones PPC. [Google Scholar]
  • Erikson, Erik. 1963. Youth, Change and Challenge. New YorkBasic Books. [Google Scholar]
  • Erikson, Erik. 1968. Identity, Youth and Crisis. New YorkW.W. Norton and Company. [Google Scholar]
  • Fons, ClaraBlanca Luque, and Maria Forteza2012“El mapa de minories religioses de Catalunya [Religious Minorities’ Map of Catalonia].” Revista Catalana de Sociologia 28: 1527. DOI: 10.2436/20.3005.01.45. [Crossref], [Google Scholar
  • Gunnoe, Marjorie L. 2000. “Predictors of Religiosity in Youth: A Longitudinal Study of the National Survey of Children.” Presented at the meeting of the Society for Research in Adolescence, Chicago, IL. [Google Scholar]
  • Helland, Christopher. 2005“Online Religion as Lived Religion.” Online-Heidelberg Journal for Religions on the Internet 1.1: 116. DOI: 10.11588/heidok.00005823. [Crossref], [Google Scholar]
  • Hemming, Peter. 2017“Childhood, Youth and Non-Religion: Towards a Social Research Agenda.” Social Compass 64(1): 113129. DOI: 10.1177/0037768616683333. [Crossref], [Web of Science ®], [Google Scholar]
  • Kim-Spoon, JugmeenGregory S. Longo, and Michael E. McCullough2012“Adolescents Who Are Less Religious than Their Parents are at Risk for Externalizing and Internalizing Symptoms: The Mediating Role of Parent-Adolescent Relationship Quality.” Journal of Family Psychology 26 (4): 636641. DOI: 10.1037/a0029176. [Crossref], [PubMed], [Web of Science ®], [Google Scholar]
  • Kozinets, Robert V. 2010. Netnography: Ethnographic Research Online. LondonSage. [Google Scholar]
  • Lea, MaryRusell, Spears, and Daphne, Groot. 2001“Knowing Me, Knowing You: Anonimity Effects on Social Identity Processes within Groups.” Personality and Social Psychology Bulletin 27: 526537. DOI: 10.1177/0146167201275002. [Crossref], [Web of Science ®], [Google Scholar]
  • Lenhart, A. 2015. Teens, Social Media & Technology Overview 2015. Pew Research Center, Internet, Science & Tech. Accessed 25 May 2016. http://www.pewinternet.org/2015/04/09/teenssocial-media-technology-2015/ [Google Scholar]
  • Leurs, Koen, and Sandra Ponzanesi2011Mediated Crossroads: Youthful Digital Diaspora. M/CJournal S.l., 142. Accessed 25 May 2016. http://journal.media-culture.org.au/index.php/mcjournal/article/view/324 [Google Scholar]
  • Leurs, Koen. 2016“Digital Divides in the Era of Widespread Internet Access: Migrant Youth Negotiating Hierarchies in Digital Culture.” In Youth 2.0, Social Media and Adolescence, edited by M. WalraveKPonnetE. VanderhovenJ. Haers and B. Segaert6178SwitzerlandSpringer. [Crossref], [Google Scholar]
  • Lipman, Matthew. 1991. Pensamiento Complejo y Educación. MadridEdiciones de la Torre. [Google Scholar]
  • Lövheim, Mia. 2015“Gendering Media, Religion and Cultura: Key Insights and New Challenges.” In Media, Religion and Gender in Europe, edited by M. Díez BoschJ. M. Carbonell, and J. L. MicóBarcelonaBlanquerna Observatory on Media, Religion and Culture. [Google Scholar]
  • Munakata, Yuko2007Information Processing Approaches to Development. Handbook of Child Psychology 2: 310. DOI: 10.1002/9780470147658.chpsy0210. [Crossref], [Google Scholar]
  • Newel, Allen, and Herbert Simon1972. Human Problem Solving. Englewood CliffsPrentice Hall. [Google Scholar]
  • Padrini, Paolo. 2015“iBreviary as New Concept of Religious App.” In Catholic Communities Online, edited by M. Díez BoschJ. M. Carbonell, and J. L. MicóBarcelonaBlanquerna Observatory on Media, Religion and Culture. [Google Scholar]
  • PapacharissiZizi, ed. 2011. A Networked Self. New YorkRoutledge. [Google Scholar]
  • Piaget, Jean. 1963. The Origins of Intelligence in Children. New YorkNorton. [Crossref], [Google Scholar]
  • DOI: 10.1177/1367549407088333. [Crossref], [Web of Science ®], [Google Scholar]
  • Zhao, ShanyangSherry Grasmuck, and Jason Martin. 2008“Identity Construction on Facebook: Digital Empowerment in Anchored Relationships.” Computers in Human Behavior, 24 (5): 18161836. DOI: 10.1016/j.chb.2008.02.012. [Crossref], [Web of Science ®], [Google Scholar]

Also discussion is poor in comparison to all the work done.

Round 2

Reviewer 3 Report

Clear improvement of the article and the suggestions. Now it has the quality for IJERPH.